# Nano-Pulse Treatment Overcomes the Immunosuppressive Tumor Microenvironment to Elicit In Situ Vaccination Protection against Breast Cancer

**DOI:** 10.3390/vaccines12060633

**Published:** 2024-06-07

**Authors:** Anthony Nanajian, Megan Scott, Niculina I. Burcus, Brittney L. Ruedlinger, Edwin A. Oshin, Stephen J. Beebe, Siqi Guo

**Affiliations:** 1Frank Reidy Research Center for Bioelectrics, Old Dominion University, Norfolk, VA 23508, USA; anthony8n@gmail.com (A.N.); m1scott@odu.edu (M.S.); nburcus@odu.edu (N.I.B.); brued@outlook.com (B.L.R.); eoshi001@odu.edu (E.A.O.); sbeebe@odu.edu (S.J.B.); 2Department of Biological Sciences, Old Dominion University, Norfolk, VA 23529, USA; 3Department of Electrical & Computer Engineering, Old Dominion University, Norfolk, VA 23529, USA

**Keywords:** nano-pulse treatment (NPT), in situ vaccination, breast cancer, tumor microenvironment, immunosuppression, apoptosis, memory T cells, regulatory T cells, myeloid-derived suppressor cells, tumor-associated macrophages

## Abstract

We previously reported that nano-pulse treatment (NPT), a pulsed power technology, resulted in 4T1-luc mammary tumor elimination and a strong in situ vaccination, thereby completely protecting tumor-free animals against a second live tumor challenge. The mechanism whereby NPT mounts effective antitumor immune responses in the 4T1 breast cancer predominantly immunosuppressive tumor microenvironment (TME) remains unanswered. In this study, orthotopic 4T1 mouse breast tumors were treated with NPT (100 ns, 50 kV/cm, 1000 pulses, 3 Hz). Blood, spleen, draining lymph nodes, and tumors were harvested at 4-h, 8-h, 1-day, 3-day, 7-day, and 3-month post-treatment intervals for the analysis of frequencies, death, and functional markers of various immune cells in addition to the suppressor function of regulatory T cells (Tregs). NPT was verified to elicit strong in situ vaccination (ISV) against breast cancer and promote both acute and long-term T cell memory. NPT abolished immunosuppressive dominance systemically and in the TME by substantially reducing Tregs, myeloid-derived suppressor cells (MDSCs), and tumor-associated macrophages (TAMs). NPT induced apoptosis in Tregs and TAMs. It also functionally diminished the Treg suppression capacity, explained by the downregulation of activation markers, particularly 4-1BB and TGFβ, and a phenotypic shift from predominantly activated (CD44^+^CD62L^−^) to naïve (CD44^−^CD62L^+^) Tregs. Importantly, NPT selectively induced apoptosis in activated Tregs and spared effector CD4^+^ and CD8^+^ T cells. These changes were followed by a concomitant rise in CD8^+^CD103^+^ tissue-resident memory T cells and TAM M1 polarization. These findings indicate that NPT effectively switches the TME and secondary lymphatic systems from an immunosuppressive to an immunostimulatory state, allowing cytotoxic T cell function and immune memory formation to eliminate cancer cells and account for the NPT in situ vaccination.

## 1. Introduction

While advances in immunotherapy, particularly checkpoint inhibitors, have been promising for breast cancer patients, clinical trials have shown that breast cancer is largely resistant to these measures [1,2,3]. One major challenge of effective immunotherapy for breast and other immunologically “cold” cancers is the immunosuppressive tumor microenvironment (TME), which is a complex network containing multiple types of immunosuppressive cells, chemokines, cytokines, physical barriers (stromal cells and extracellular matrix), chemical factors, metabolites, etc. [4,5]. Various approaches and strategies have been proposed and studied to target these TME immunosuppressive components and improve treatment outcomes [6,7].

In situ vaccination (ISV) is a 130-year-old concept that has had some efficacy but with initial controversies. The original approach came from reports by Dr. William Coley in the 1890s using microorganisms, a mixture of *Streptococcus pyogenes* and *Serratia marcescens*, to treat unresectable sarcoma. Although Coley developed a variety of strategies, the approach was not widely accepted because minimal quality control of the microbial reagent caused a lack of reproducibility. In 1957, a study [8] showed that tumors could be recognized by the immune system, beginning a slow increase in tumor immunotherapy research. In 1959, Bacillus Calmette–Guerin (BCG) was the first reported ISV approach to treat mouse fibrosarcoma [9]. Later, clinical studies demonstrated its efficacy and safety in bladder cancer, which led to the FDA approval of the first cancer immunotherapy in history. The BCG approach presented no cancer antigens but activated the immune system. In addition to BCG, toll-like receptor (TLR) agonists, oncolytic viruses, cytokines, radiation, and hypothermia [10], emerging novel drugs/technologies, including checkpoint inhibitors, nanoparticles, electrochemotherapy (ECT) [11,12], gene electro-transfer (GET) [13,14], and nano-pulse treatment (NPT) [15,16,17], can serve as ISV approaches as well. The most documented but rare phenomenon is what radiation oncologists term the abscopal effect. In these rare cases, irradiation shrinks one localized tumor and then shrinks tumors outside the irradiated zone. Radiation damage to tumors can therefore stimulate systemic antitumor immunity, leading to the regression of metastatic cancer. Like radiation, moderate hyperthermia with iron oxide nanoparticles and an alternating magnetic field induces local hyperthermia to damage tumors, demonstrating the promising potential of local hyperthermia treatment to induce antitumor immune responses [18].

ISV has attracted increasing attention as a potential paradigm shift for cancer immunotherapy [10,19]. Unlike conventional vaccines or systemic immunotherapies, ISV can overcome the immunosuppressive TME through the direct administration of drugs/adjuvants into the tumor. Current advances in tumor biology, immunology, technological innovations, and targeted therapies have improved immune outcomes of ISV approaches. Nano-pulse treatment (NPT), also known as nano-pulse stimulation (NPS) or nanosecond pulsed electric fields or electric pulses (nsPEFs/nsEPs), is a pulsed-power technology that compresses electric energy and releases it in high-powered (generally 100 megawatts) nanosecond (1~999 ns)-duration electric pulses [20,21]. NPT has been demonstrated to effectively ablate tumors in multiple animal models [15,22,23,24,25]. In our previous studies, NPT ablated localized tumors in poorly immunogenic cancer models, including orthotopic 4T1-luc mouse triple-negative breast cancer [15], N1S1 rat liver models, and ectopic [24] Pan02 mouse pancreatic cancer [17]. Importantly, after the complete regression of primary tumors, NPT resulted in a 75–100% rejection of secondary live tumor challenges in these three cancer models, demonstrating its strong ISV effects. The 4T1 tumor model is a very aggressive and spontaneously metastatic malignancy with abundant immune suppressive cells in the TME, including regulatory T cells (Tregs), tumor-associated macrophages (TAMs), and myeloid-derived suppressor cells (MDSCs) [26,27]. The mechanisms behind the NPT-induced ISV effect are not well understood. Considering immunosuppressive prevalence in the TME, the immune system must overcome these roadblocks prior to the generation of antitumor immunity. To further clarify how NPT counteracts the immunosuppressive TME to elicit a strong ISV effect, in this study, we investigated the dynamic changes in major immunosuppressive cells (Tregs, MDSCs, and TAMs), the Treg suppressive activity, and the acute and long-term T cell immune memory formation.

NPT is a novel type of drug-free ISV approach that can ablate primary tumors to generate strong vaccine protection against the second live tumor challenge. In the breast cancer model, the ISV effect correlated with an acute rise in tissue-resident memory T cells and the formation of long-term T cell memory. NPT selectively induced activated Treg cell apoptosis and reduced their functional suppressive capacity while preserving cytotoxic T cells. NPT also eradicated TAMs by directly killing them and causing late diminishment of MDSCs without evidence of the induced cell death mechanism.

## 2. Materials and Methods

### 2.1. Animals and Cell Lines

Female BALB/c mice (8–10 weeks of age) were purchased from Jackson Laboratory. The mice were housed and maintained at the ODU AAALAC-accredited animal facility. All animal procedures in this study were approved by IACUC at Old Dominion University.

The 4T1-luc cells were provided by Dr. G Gary Sahagian at Tufts University and were maintained in high glucose DMEM (ATCC^®^30-2002^TM^) (ATCC, Manassas, VA, USA) supplemented with 10% FBS, NEAA, and antibiotics (100 units/mL penicillin and 100 μg/mL streptomycin). The 4T1-luc cells with passage numbers 2–7 were thawed for expansion. Cells with passage numbers 9–20 were used in the described experiments. Cells were tested with MycoAlert^®^ Mycoplasma Detection Kits (Lonza, Morristown, NJ, USA) to exclude mycoplasma contamination.

### 2.2. In Vivo Nano-Pulse Treatment and the Secondary Live Tumor Challenge

Tumors were initiated by injecting female BALB/c mice with 1 × 10^6^ 4T1-luc cells in 50 μL DPBS (Life Technologies, Carlsbad, CA, USA) in the left posterior mammary fat pad. The control group received the tumor inoculation only. The remaining mice underwent NPT on Day 11 following tumor inoculation (Appendix A), when average tumor sizes were 6–8 mm or 60–100 mm^3^. In all studies, the NPT parameters included a pulse duration of 100 ns, electric field strengths of 40–50 kV/cm, 1000 pulses, and 3 Hz. Nanosecond electric pulses were delivered to the tumor tissue using a two-plate pinch electrode with an 8 mm diameter. Prior to treatment, all hair was thoroughly removed, and the injection site was briefly treated with Nair. The tumor mass and electrodes were covered with ultrasound gel to maintain good tumor–electrode contact and prevent air accumulation, which creates electrical breakdowns.

Animals who were tumor-free over 7 weeks were challenged orthotopically in the right posterior mammary fat pad with 0.5 × 10^6^ live 4T1-luc tumor cells. Tumor growth was monitored twice weekly by caliper measurements.

### 2.3. Tissue Harvesting and Processing for the Analysis of Immune Cells

The treated mice were euthanized 4 hour (4 h), 8 hour (8 h) (for local tissues only), and on Day 1 (D1), Day 3 (D3), and Day 7 (D7) post-NPT (Appendix A). Their tumor tissues, tumor-draining lymph nodes (dLNs), blood, and spleens were harvested. Control tissues were obtained from mice with untreated tumors. Single-cell suspensions from each tissue were prepared to analyze immune cells, including CD3, CD4, CD8, and tissue-resident marker CD103. To examine the effect on memory and central memory T cells, 3 months post-NPT, tumor-free animals were euthanized. Spleens and blood were harvested. Spleens/blood of tumor-bearing mice were used as control. Single-cell suspensions were prepared from spleens or blood and then stained with CD3, CD4, CD8, CD44, and CD62L antibodies.

To quantify IFN-γ production and IFN-γ producing T cells, splenocytes (2 × 10^6^/mL), 1 mL per well, were incubated with media and tumor lysate (10 µg/mL) or plate-bound low-endotoxin/azide-free LEAF anti-CD3 Ab (0.5 µg/mL in DPBS) in a 24-well plate. For intracellular cytokine staining, cells were incubated for 6 h, and Monensin added for the final 4 h. For IFN-γ production, cells were incubated for 24 h, and supernatants were collected for the ELISA assay (Biolegend, San Diego, CA, USA).

### 2.4. Preparation for Single-Cell Suspensions

The primary solid tumor, dLN, blood, and spleen were collected and prepared into single-cell suspensions for downstream analysis. The spleen and dLN were gently mashed through a 70 µm cell strainer into a conical tube. The dissociated spleen and the blood underwent RBC lysis to make cell suspensions for further analysis. For solid tumor processing, the fat and other surrounding tissues were removed. The tumor samples were washed with RPMI 1640 (Life Technologies) and cut into 1–5 mm^3^ pieces. The cut pieces were then dissociated with the Miltenyi Biotec tumor dissociation kit and the Gentlemacs Octo-dissociator (with heater) using the tough tumor dissociation protocol. The digest was then passed through a 70 µm cell strainer to remove clumps. Tumor single-cell suspensions then underwent magnetic bead-based CD45 TIL isolation (Miltenyi Biotec, Gaithersburg, MD, USA) to obtain CD45^+^ cells for downstream flow cytometric analysis.

### 2.5. Flow Cytometry

To perform cell surface staining, single-cell suspensions were washed with FACS buffer containing PBS with 2% FBS (Life Technologies) and then incubated with anti-FcR (TruStain FcX™ PLUS, Biolegend) for 10 min on ice to block unspecific binding of antibodies. The following fluorochrome-labeled antibodies were used for Treg, MDSC, and TAM labeling: CD4 FITC, CD8 APC/Cy7, CD3 BV510, CD25 APC, CTLA-4 PerCP/Cy5.5, PD-1 PE/Cy7, CCR4 BV421, 4-1BB APC, TGFβ BV421, CD45 Pacific Blue, CD11b PE, Gr-1 PE/Cy7, NKp46 APC, F4-80 FITC, and CD86 APC/Cy7. All cell surface antibodies were purchased from Biolegend.

For intracellular and intranuclear staining, single-cell suspensions were first labeled with cell surface antibodies, followed by fixation and permeabilization using the Foxp3 transcription factor buffer set (Thermofisher Scientific). Permeabilized cells were then labeled with Foxp3 PE, Helios PerCP/Cy5.5, IL-17 PE/Cy7, RORγt PerCP-Cy5.5, and/or IFN-γ PerCP-Cy5.5 primary antibodies (Thermofisher Scientific). Cytokine staining using the above buffer and cytokine antibodies from Thermofisher Scientific was verified with the company’s in-house data as well as our own experimental data. This technique allowed us to co-stain for Foxp3, RORγt, and IL-17 in the same panel to investigate T cell reprogramming.

A sample Treg gating strategy is shown (Appendix A), whereby Tregs are identified as CD3^+^CD8^−^CD4^+^CD25^+^Foxp3^+^ T cells. T effector and Tconv refer to non-Treg CD4 (CD3^+^CD8^−^CD4^+^CD25^−^Foxp3^−^) and CD8 (CD3^+^CD4^−^CD8^+^CD25^−^Foxp3^−^) T cells, respectively.

For cell apoptosis studies, cells were labeled with Zombie NIR and Annexin V (Biolegend). Due to the sensitivity of the phosphatidylserine bond (for Annexin V binding) to the fixation/permeabilization process (for Foxp3 analysis), we did not incorporate Annexin V and Foxp3 co-staining in the same panel. To perform the cell apoptosis studies, freshly obtained single-cell suspensions were first labeled with Zombie NIR using a serum-free PBS buffer. Cells were then washed with FACS buffer to perform cell surface staining. The labeled cells were then washed and resuspended in Annexin V binding buffer, stained with Annexin V, and immediately analyzed by flow cytometry.

Stained cells were analyzed on a MACSQuant 10 Analyzer, BD FACS Calibur, or BD FACS Canto II at Old Dominion University and Eastern Virginia Medical School. The acquired data were analyzed using FlowJo V10 software.

### 2.6. In Vitro Treg Suppression Assay

CD4^+^CD25^+^ Tregs were isolated from dLN of tumor-bearing mice and NP-treated mice on post-treatment Day 2. To obtain a sufficient quantity of Tregs, dLN from 2–3 mice were pooled together for each group. Spleen-derived CD8 responder (Tresp) cells were isolated from naïve mice by negative selection using magnetic beads (Stemcell Technologies). Purified responder cells were labeled with 5 µM CFSE (ThermoFisher Scientific) and plated in a 96-well round-bottom plate at a density of 4 × 10^4^ responder cells per well. CD4^+^CD25^+^ Tregs were co-incubated with CFSE-labeled responder cells at the Treg/Tresp ratios 1:1, 1:2, and 1:4. The plates were incubated at 37 °C with 5% CO_2_ for 60 h in the presence of CD3/CD28 activation beads (ThermoFisher Scientific). Responder cell proliferation was quantified by flow cytometry based on the dilution of the CFSE dye. Treg suppression was calculated in the following manner: %Suppression = [1 − (%proliferating Tresp at Treg/Tresp ratio/%proliferating Tresp-only cells)] ×100.

### 2.7. Statistical Analysis

All values were reported as the mean ± SD. Analyses of quantitative data, including CD4^+/^CD8^+^ T cells, MDSCs, Tregs, TAMs, and their phenotypic, functional, and/or death markers were performed by one-way ANOVA for multiple time-point comparisons or *t*-test between two groups with a minimum sample size of 3 mice per group. Animal survival was assessed with Kaplan–Meier Survival Analysis (LogRank test). The rate of ISV protection against a second live tumor challenge was analyzed with chi-square test. Results were considered statistically significant at *p*-values less than 0.05. Statistical analyses were carried out using Vassarstats (http://vassarstats.net/, accessed on 16 May 2020) and SigmaPlot 12.5 (Systat Software, Inc., Chicago, IL, USA).

## 3. Results

### 3.1. NPT Elicits Strong In Situ Vaccination (ISV) Protection

Following NPT of the primary tumors, mice that underwent complete tumor regression over 7 weeks were then challenged with a second live tumor (Figure 1A). As shown in Figure 1B, a single NPT elicited a strong ISV effect, with 81.5% (22/27) of the animals protected against a second live tumor challenge. This result was consistent with our previous report that a single NPT achieved 100% (11/11) ISV protection in the same 4T1 model [15]. In contrast, none of the age-matched naïve mice (0/14) were able to reject the secondary tumor challenges. Noticeably, even though five (18.5%) animals treated with NPT failed to reject the live tumor challenge, these animals still gained an additional 10 ~ 14-day survival in comparison to control animals (Figure 1B). This occurrence is likely due to slow or delayed tumor growth in these animals during the initial 2–3 weeks following the second tumor challenge.

### 3.2. NPT Induces Antitumor Immune Memory Responses

To characterize early antitumor immune responses, the emergence of tissue-resident memory T cells (Trms), the major cytotoxic cell type for eliminating tumors, was examined. Following NPT, CD103^+^CD8^+^ Trm numbers (Figure 2A) were continuously elevated throughout the first week. The total cell counts significantly increased by 1.9- and 2.4-fold in the dLN by Day 3 and Day 7, respectively. Consistent with this initial acute local T cell response, the frequencies of both CD4^+^ and CD8^+^ memory T cells, both effector memory (Tem, CD44^+^CD62L^−^) and central memory (Tcm, CD44^+^CD62L^+^) T cells, subsequently increased over three months post-treatment. Notably, the frequencies of CD8^+^ Tcms and Tems were elevated up to 85- and 10-fold in the blood and 71- and 8-fold in the spleens of NP-treated animals, respectively. This is in contrast to untreated tumor-bearing animals (Figure 2B). Moreover, the numbers of functional cytotoxic T cells or IFN-γ^+^ CD8^+^ T cells in the spleens were greatly increased, up to 13-fold compared to levels in untreated tumor-bearing mice (Figure 2C). The frequencies of CD4^+^ Tcms and Tems were elevated up to 17- and 4-fold in the blood and 15- and 4-fold in the spleens of NP-treated animals, respectively, in contrast to the untreated animals (Figure 2D). Similarly, IFN-γ^+^ CD4^+^ T cells in spleens were significantly increased up to nine-fold compared to the levels in untreated tumor-bearing mice (Figure 2E). Importantly, these memory T cells in spleens also exhibited superior antitumor immunity, indicated by the significantly higher production of IFN-γ. The levels of secreted IFN-γ were significantly increased, at 6- and 58-fold in the splenocytes of NP-treated mice, in response to co-culturing with tumor lysate and CD3 activation, respectively, in contrast to those of untreated tumor-bearing mice (Figure 2F).

Considering the immunosuppressive dominance in the 4T1 tumor [28], we next sought to investigate how NPT elicited a strong ISV effect and antitumor immune memory in such an environment. Therefore, the major immunosuppressive cells in the blood, spleen, dLN, and TME were examined.

### 3.3. NPT Overturns the Treg Dominance in the TME and Systemically

Treg frequency among CD4 T cells was assessed in the blood (Figure 3A,B), spleen (Figure 3C), dLN (Figure 3D), and TME (Figure 3F). In untreated mice, the Treg percentage among CD4 T cells was 18.6% in blood, 23.8% in spleen, 20.6% in dLN, and 49.5% in tumor tissues. After 4T1 tumor-bearing mice were treated with 100 ns NPT, a significant and sustained drop in Tregs was observed both systemically and locally. The frequencies of Tregs were decreased post-treatment by as much as 52.1% in the TME (4 h), 42.2% in the dLN (Day 3), 52.3% in the blood (Day 3), and 19.7% in the spleen (Day 7) (Figure 3A–F). The Treg cell count decreased 4.4-fold by 4 h and 8.1-fold by Day 3 from its untreated intratumoral baseline (Figure 3G). Tconv, in contrast, only decreased 1.4-fold by 4 h and 2.3-fold from the baseline by Day 3 post-treatment. These changes shifted the ratio of intratumoral Treg vs. Tconv from 1:1 in untreated tumors to 1:3 by 4 h and 2:7 by Day 3 post-treatment (Figure 3G).

### 3.4. NPT Selectively Eradicates Tregs but Spares CD8 and CD4 Tconv Cells

To understand why NPT resulted in a remarkable decrease in Tregs but a less pronounced decrease in Tconv (Figure 3E,F), whether cell death occurred differentially in various subsets of T cells was determined. As shown in Figure 4A–C, both activated (CD44^+^CD62L^−^) and naïve (CD44^−^CD62L^+^) Treg subsets, were the only T cell populations to exhibit statistically significant increases in apoptotic changes. Activated Tregs exhibited the highest level of apoptosis, seen at 4 h post-treatment (Figure 4B,D). Importantly, CD4 Tconv and CD8 T cells showed little or no change in apoptosis following NPT (Figure 4C).

Noticeably, all T cell groups, except for the naïve Treg subset, exhibited various levels of apoptotic death in the control mice (Figure 4C). Therefore, the percent changes among the T cell subsets from their baseline apoptotic status to post-NPT apoptosis at 4 h post-treatment were evaluated. Changes in T cell subset apoptosis were calculated as follows: (% apoptosis at 4 h post-treatment − % apoptosis in untreated control sample)/% apoptosis in untreated control sample. Tregs exhibited the highest change in apoptosis at 4 h post-NPT. Activated Tregs, in particular, exhibited a 53.7% increase in apoptosis (Figure 4D), whereas CD8 T cells showed no change and CD4 Tconv had a small but insignificant increase in apoptotic death.

### 3.5. NPT Reduces Treg Suppression Capacity

In vitro Treg suppression assays were conducted to assess if NPT alters the Treg suppression function. Tregs were isolated and incubated with CFSE-labeled CD8 T cells at different Treg/CD8 ratios in the presence of activation beads. Tregs isolated from dLNs of NP-treated mice exhibited a 50.5% reduced suppression capacity compared to those from untreated tumors (control group) at a 1:1 Treg/CD8 co-culture ratio (Figure 5A). To further explore why the Treg function was impaired, we next examined if various activation markers were altered by NPT. As seen in Figure 5B, the activated (CD44^+^CD62L^−^) Tregs were the initially dominant Treg population in the dLN of tumor-bearing mice. These activated Tregs outnumbered naïve Tregs at a 2:1 ratio. Following NPT, the activated/naïve (CD44^−^CD62L^+^) Treg ratio shifted to 1:1 by Day 1, and naïve Tregs became the dominant Treg population by post-treatment Day 3, at an activated/naïve Treg ratio of 1:2 (Figure 5B,D).

The T cell activation marker 4-1BB, which has significant immunotherapeutic potential, was expressed among Tregs in the dLN (Figure 5C,E), tumors (Appendix A), and spleens (Figure 5F) of tumor-bearing mice. However, 4-1BB expression was not found among (a) Tregs in the blood, (b) Tregs in any healthy (naïve) mouse tissues (Appendix A, negative control), or (c) in conventional CD4 (Appendix A) or CD8 T cells of healthy or tumor-bearing mice. A significant reduction in the total number of 4-1BB^+^ Tregs in the dLN and spleen was observed on Days 1–7 following NPT. Day 1 post-NPT had the most pronounced reduction, with a greater than three-fold reduction in 4-1BB^+^ Tregs in both the dLN and spleen when compared to untreated tumor-bearing mice (Figure 5E,F). Interestingly, there was an early transient peak among 4-1BB^+^ Tregs, around a two-fold increase from baseline, by 4 h post-NPT in the dLN, followed by a seven-fold drop by 24 h post-treatment (Figure 5C,E).

The 4-1BB^+^ Tregs were exclusively present among activated Tregs, and they were mostly absent among naïve Tregs in the dLN of tumor-bearing mice (Appendix A). A higher TGFβ expression was also found among 4-1BB^+^ and activated Tregs (Appendix A) when compared to their 4-1BB^−^ and naïve Treg counterparts. Similar to the changes in 4-1BB expression, this elevated TGFβ expression among dLN Tregs had a transient increase at 4 h post-NPT, followed by a decline in the days following treatment (Figure 5G and Appendix A). While we could clearly observe the above phenotypic changes among dLN Tregs, the few remaining intratumoral Tregs post-NPT made it challenging to properly describe such phenotypic changes among TILs (Appendix A).

### 3.6. NPT Decreases MDSCs and TAMs with Distinctive Dynamics and Mechanisms

While investigating the intratumoral immune cell changes, backgating was performed to study the distribution of intratumoral TAMs (CD11b^+^ F4/80^+^), MDSCs (CD11b^+^ Gr-1^+^), and CD3^+^ T lymphocytes on the forward vs. side scatter plot, and it was found that NPT affected these cell populations differently. TAMs showed a rapid decrease in cell count (by Day 1 post-treatment) (Figure 6A), which was associated with a significant increase in apoptosis as early as 4 h post-treatment (Figure 6B) and remained below control levels on Day 3 post-treatment (Figure 6B). In contrast, MDSC counts remained at control levels one day after treatment but decreased by Day 3 (Figure 6A). Notably, a 40% decrease in MDSC death markers was observed in the TME at 4 h post-NPT (Figure 6C). To further classify TAMs as M1- and M2-phenotypes, the expressions of CD80, CD86, and MHC-II were examined. Both CD86 and MHC-II expressions were significantly upregulated on the TAMs at 1–3 days post-NPT (Figure 6D).

Additionally, the impact of NPT on TAMs in the dLN and MDSCs in blood was assessed. TAMs in the dLN of untreated tumor-bearing mice made up 5.06% of the total CD45^+^ leukocyte population (Appendix A), and MDSCs made up 67.7% of the total blood leukocyte population (Appendix A). By Day 3 post-NPT, TAMs in the dLN and MDSC populations in the blood were decreased by 90% and 75%, respectively (Appendix A). Noticeably, MDSCs made up a very small fraction of the dLN total leukocyte population (around 0.10%) and declined to a negligible value (less than 0.02%) following NPT.

## 4. Discussion

Mechanistically, there are two aspects for optimal ISV: (1) the induction of the immunogenic cell death of cancer cells, and (2) the switch of an immunosuppressive TME to an immunostimulatory environment, both of which are fulfilled by NPT, while the other strategies often require two different treatments to embrace each attribute. Regarding the first aspect, our study reported the regression of Tregs and TAMs by apoptosis and MDSC by another cell death mechanism. This event provided an environment for the simultaneous rise in CD8^+^CD103^+^ tissue-resident memory T cells and TAM M1 polarization. The second independent mechanism is the production of immunogenic cell death (ICD) molecules. NPT induced the release of ATP, calreticulin, and HMGB [15]. Our observations support NPT as a novel type of ISV approach. However, whether NPT can modify the TME, especially whether it can directly impact immune cells and their function, has not been previously explored. Our research discoveries in this study provide further insight into NPT-induced antitumor immunity, specifically its profound impact on the TME immunosuppressive cells in the 4T1-luc model.

NPT appears to collapse all three major immunosuppressive cell populations in the TME. Interestingly, the distinctive cell frequency and death marker changes among each group hint at the involvement of population-specific intrinsic mechanisms. A rapid drop in cell counts concomitant with the presence of apoptotic death markers as early as 4 h post-NPT occurs in both Tregs and TAMs. These results indicate that NPT likely induces cell death or directly kills these suppressive cells. However, a delayed decrease in cell counts takes place in MDSCs on Day 3 post-NPT. This result, together with a reduction in apoptotic markers 4 h post-treatment, suggests an alternative mechanism behind how NPT diminishes MDSCs. Considering the increase in dendritic cells in the TME previously reported by our group in both 4T1 breast [15] and Pan02 pancreatic [22] cancer models, we suspect that NPT may shrink the MDSC population not by direct induction of cell death but rather by promotion of MDSC differentiation into dendritic cells (DCs). The differentiated DCs can further participate in antigen presentation and promote antitumor T cell response. This postulate needs further investigation.

A significant discovery in this study is that NPT exhibits selective cytotoxicity towards Tregs while sparing effector CD4^+^ and CD8^+^ T cells. Treg predominance suggests worse outcomes for many cancer types, including breast, while a high CD8^+^ or CD8^+^/Treg ratio is associated with a better prognosis [29,30,31,32]. A significant knockdown of Tregs and an increase in effector CD4^+^ and CD8^+^/Treg ratios would favor the induction of antitumor immunity. Prior studies by Plaza-Sirvent et al. may help clarify the above selective cytotoxicity observation. The group demonstrated that Tregs had an increased sensitivity to apoptosis compared to Tconv [33]. This sensitivity was correlated with a lower expression of c-Flip_L_, a member of the extrinsic apoptosis family, among Tregs when compared to Tconv. Among Tregs, the increased sensitivity to apoptosis was mostly restricted to activated CD44^+^CD62L^−^ Tregs when compared to naïve CD44^−^CD62L^+^ Tregs [33]. Mirandola et al. reported that both resting and activated CD8^+^ T cells showed a high expression of c-FLIPs as well [34]. These studies are consistent with our findings that the ratios of activated Tregs (CD44^+^CD62L^−^) to naïve Tregs (CD44^−^CD62L^+^) and Tregs to CD8^+^ T cells significantly decreased on Days 1 and 3 after NPT. The lower c-Flip_L_ expression in activated Tregs may explain why they are more vulnerable than their Tconv/CD8^+^ counterparts to apoptotic changes resulting from NPT. Our results also support a proposal by Drs. Overacre-Delgoffe and Vignali, who suggested that Treg fragility may be a valuable target for effective antitumor immunity [35].

Another important finding is the ability of NPT to impair the Treg suppressive function, likely via the selective removal of the activated Treg (CD44^+^CD62L^−^) phenotype discussed above. The analysis of functional markers, including 4-1BB and TGFβ expression, further supports this concept. The expressions of both 4-1BB and TGFβ are dramatically diminished by 1-day post-NPT and remain at low levels for at least one week. These findings show a distinct contrast to results from radiotherapy, which has been studied as an ISV or enhancement for immune outcomes. Radiotherapy has been demonstrated to not only increase the frequency of Tregs by promoting their proliferation but also amplify their functional capacity [36,37,38,39]. Noticeably, the dynamics of Tregs are very different between radiotherapy and NPT. NPT rapidly shrinks Tregs as early as 4 h post-treatment and sustains the low level for at least one week, whereas radiotherapy expands Tregs 3 days after treatment and then maintains the high level for 9–13 days [37,39]. The mechanisms behind these distinct Treg responses between NPT and radiotherapy warrant further exploration.

We also note that 4-1BB is exclusively expressed among Foxp3^+^ CD4 Tregs in tumor-bearing mice, is absent among other T cell subsets, and is entirely absent in naïve mouse T cells. The expression of 4-1BB on Tregs is also downregulated post-treatment, making it an excellent marker to track the impact of NPT on local and systemic Treg dynamics. Additionally, studies suggest that 4-1BB is a valuable therapeutic target to enhance antitumor immunity [40,41]. Therapies targeting 4-1BB can greatly improve the efficacy of immunotherapy via the depletion of Tregs [40]. The depletion of 4-1BB^+^ Tregs can inhibit tumor growth, and Tregs lacking 4-1BB exhibit impaired suppressive function [41]. Although no systemic antibody is administered in our study, the depletion of 4-1BB^+^ Tregs can still be achieved by NPT. This non-drug approach to selectively deplete active Tregs has great potential for cancer immunotherapy because it should minimize immune-related adverse events commonly associated with systemic immunotherapy.

As the TME immunosuppressive prevalence is being removed, the initiation of antitumor immune responses is evidenced by rapid DC activation, TAM-M1 polarization, and CD8^+^ Trm elevation. Thus, NPT encourages coordinated host changes in both the lymphoid and myeloid immune systems to shift from an immunosuppressive to an immunostimulatory performance as part of NPT-induced immune responses. The overall immunomodulation of NPT in the TME is shown in Figure 7. Consequently, the dramatic increase in long-term effector and central memory T cells and their cytotoxic function is consistent with a strong ISV effect whereby the complete rejection (or delayed tumor growth) of the secondary tumor challenge is achieved following the successful treatment of the primary tumor.

While this study did not examine other immunosuppressive cells, such as tumor-associated fibroblasts, tumor-associated neutrophils, regulatory B cells, etc., the reversal from an immunosuppressive to an immunostimulatory TME in this model supports an ISV-induced shift in immunity. Future topics of investigation include the following: (1) the characterization of other immunosuppressive cells, and (2) whether the above immunosuppressive reversal and the ISV induction are NPT-parameter dependent. So far, our available data imply that NPT with short durations, fast rising times, and high electric field strengths seem better than those with long pulse durations and low electric field strengths. NPT with 100 ns and 40–50 kV/cm was reported to result in 75–100% ISV protection in Pan02 pancreatic [17], 4T1 breast [15], and N1S1 liver cancer [16] models, whereas NPT with 200 ns and 25–30 kV/cm only led to 33.3% and 0% ISV protection in B16 melanoma [42] and Pan02 pancreatic cancer [22] models, respectively. Nevertheless, whether the NPT-induced ISV is tumor-intrinsic or NPT-parameter-dependent can be determined by utilizing the same cancer model treated with various sets of NPT parameters. Our group is also investigating the molecular mechanisms of NPT-selective toxicity to Tregs over effector T cells and how NPT diminishes MDSCs without the induction of extra cell death.

## 5. Conclusions

We demonstrated that NPT, a pulsed-power technology, can be utilized as a non-drug electrical ISV approach for the ablation of primary tumors to induce a strong T cell immune memory response and protect animals against secondary tumor challenges in a poorly immunogenic mouse breast cancer model. NPT inverted the TME from immunosuppressive to immunostimulatory status via the destruction of/reduction in Tregs, TAMs, and MDSCs with the preservation of effector T cells and the promotion of TAM-M1 polarization and tissue-resident memory T cells. NPT undermined the suppressor function of Tregs and selectively eradicated activated Tregs. Consequently, long-term antitumor T cell memory was established to achieve ISV protection. These findings add pulsed-power technology as a novel ISV strategy, which makes it stand out among other ISV agents and modalities.

## Figures and Tables

**Figure 1 vaccines-12-00633-f001:**
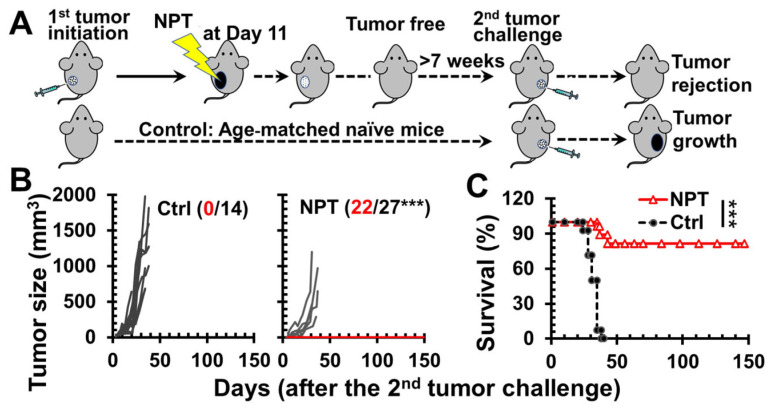
NPT tumor ablation results in in situ vaccination protection. (**A**) NPT and tumor re-challenging scheme. Female BALB/c mice with orthotopic tumors (6–8 mm) are treated with NPT. Animals tumor-free over 7 weeks are challenged orthotopically (4T1-luc) with 0.5 × 10^6^ live tumor cells. Ctrl: age-matched naïve mice without prior NPT. NPT: mice tumor-free over 7 weeks after NPT (100 ns, 40–50 kV/cm, 3 Hz, and 1000 pulses) are rechallenged with live tumor cells. (**B**,**C**) 4T1 tumor growth and survival curves of animals following a second live tumor challenge: the number of tumor-free (red) vs. total mice is indicated. ***: *p* < 0.001.

**Figure 2 vaccines-12-00633-f002:**
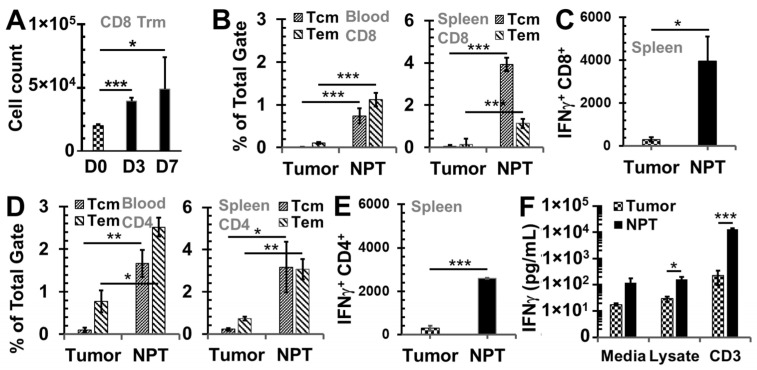
NPT elicits antitumor memory response. (**A**) Cell counts of CD8^+^ Trms per million CD45^+^ cells in dLNs. D0, D3, and D7: Days 0, 3, and 7. (**B**,**D**) CD8^+^ (**B**) and CD4^+^ (**D**) memory T cells in the blood and spleens of mice. Tcm: CD44^+^CD62L^+^ T cells; Tem: CD44^+^CD62L^−^ T cells. (**C**,**E**) IFN-γ^+^ CD8^+^ (**C**) and CD4^+^ (**E**) T cells from splenocytes after 6 h incubations with plate-bound anti-CD3. (**F**) IFN-γ product of splenocytes after 24 h incubation with tumor lysate. Groups: Tumor: untreated tumor-bearing mice, and NPT: NP-treated mice. Note: in B-F, tumor mice were at the end point for euthanasia (age 15–16 weeks), while NPT mice were euthanized post-treatment at 3 months (age 21–22 weeks). *N* = 3–5. Error bars: SD. *: *p* < 0.05, **: *p* < 0.01, and ***: *p* < 0.001 (one-way ANOVA or *t*-test).

**Figure 3 vaccines-12-00633-f003:**
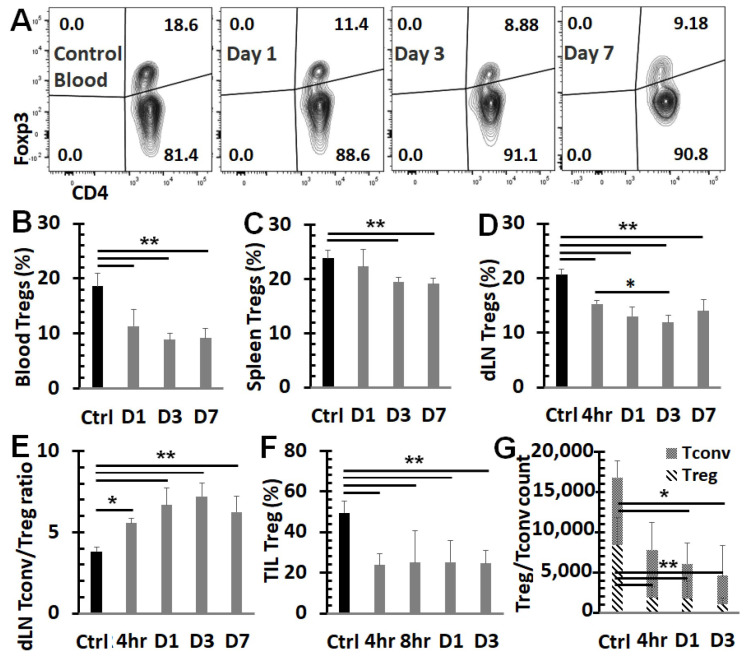
NPT reverses the Treg dominance locally and systemically. Breast tumors were established with an injection of 1 × 10^6^ 4T1-luc cells into the posterior part of the mammary fat pad. The control group (*n* = 4) received the tumor inoculation only. The remaining mice underwent NPT (100 ns pulses, 50 kV/cm, 3 Hz, 1000 pulses) on Day 11 following tumor inoculation. The treated mice were euthanized 4 h (4 h), 8 h (8 h), and on Day 1 (D1), Day 3 (D3), and Day 7 (D7) post-NPT. Their tumor tissues, tumor-draining lymph nodes (dLNs), blood, and spleens were harvested. Control tissues were obtained from mice with untreated tumors. (**A**) Summary flow plots represent Foxp3^+^ Tregs and Foxp3^− _^ Tconv among the total CD4^+^ T cell population in the blood. (**B**–**D**) Quantitative bar graphs depict the percentage of Tregs among the CD4^+^ T cell population in blood (**B**), spleens (**C**), or dLNs (**D**). (**E**) The Tconv/Treg ratios dLNs. (**F**) TIL Tregs are represented in quantitative bar graphs as the percentage of Tregs among CD4^+^ TILs. (**G**) A standardized CD4 Treg vs. CD4 Tconv cell count. *N* = 4 per group. Error bars, SD. **: *p* < 0.01 and *: *p* < 0.05 (one-way ANOVA).

**Figure 4 vaccines-12-00633-f004:**
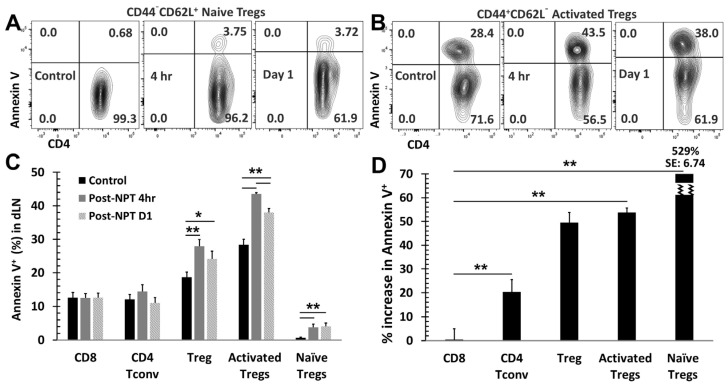
Changes in apoptosis among T cell subsets following NPT. (**A**,**B**) Summary flow plots represent Annexin V expression among activated and naïve Treg subsets in the dLN at 4 h and 1-day post-NPT. (**C**) Quantitative graph shows Annexin V expression among CD8, CD4 Tconv, CD4 total Treg, activated Treg, and naïve Treg subsets in the dLN at 4 h and 1 day post-NPT. *N* = 4 per group. Error bars, SD. (**D**) Quantitative graph shows the percentage of Annexin V expression increase, among the above subsets, from the untreated control to 4 h post-treatment. Error bars, SE. *: *p* < 0.05 and **: *p* < 0.01 (one-way ANOVA).

**Figure 5 vaccines-12-00633-f005:**
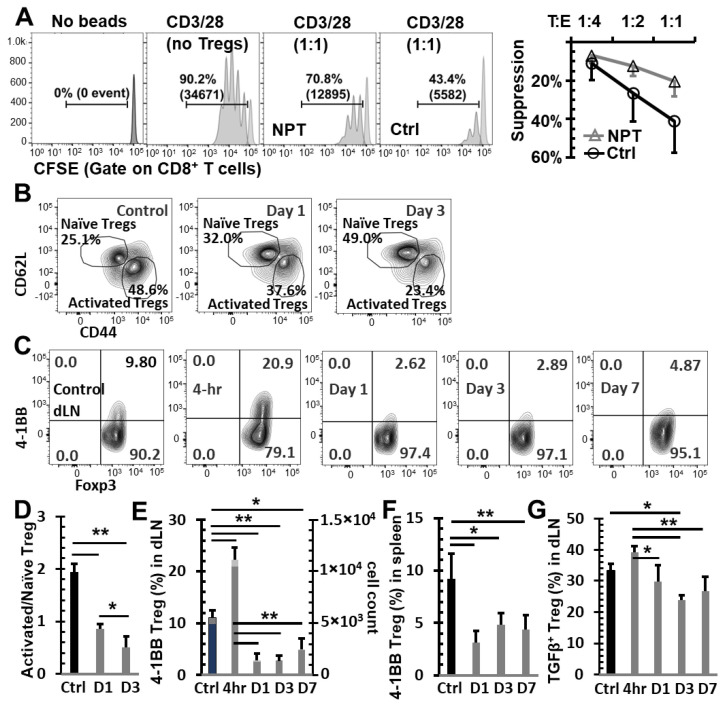
NPT eradicates activated Tregs and impairs Treg function. (**A**) In vitro suppression assay showed a reduced functional suppression capacity of Tregs isolated from the NP-treated mice. Tregs isolated from dLNs of tumor-bearing (Ctrl) or NP-treated (NPT) mice were incubated with CFSE-labeled responder cells at the Treg/Tresponder ratios of 1:1, 1:2, and 1:4 for 60 h in the presence of CD3/CD28 activation beads. Responder cell proliferation was analyzed based on the dilution of the CFSE dye. The quantitative plots represent the percentage suppression at each Treg/Tresponder ratio in the control and treatment groups. (**B**,**D**) Changes in activated and naïve Treg distribution in the dLN are represented in the summary flow plots (**B**) and quantitative bar graphs (**D**). (**C**,**E**,**F**) Phenotypic changes in the 4-1BB activation marker expression among Foxp3^+^ Tregs are represented in the summary flow plots (**C**) and quantitative bar graphs (**E**,**F**) in the dLN (**C**,**E**) and spleen (**F**). (**G**) Changes in TGFβ expression among Tregs in the dLN are represented in the quantitative bar graph. *N* = 4 per group. Error bars, SD. *: *p* < 0.05 and **: *p* < 0.01 (one-way ANOVA).

**Figure 6 vaccines-12-00633-f006:**
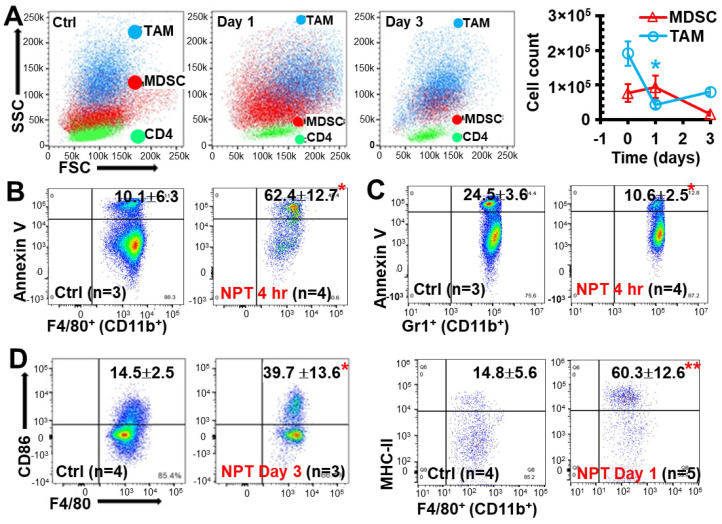
NPT diminishes intratumoral TAMs and MDSCs with differential characteristics. (**A**) Changes in intratumoral TAMs and MDSCs distribution on Day 1 and Day 3 post-treatment are shown in a representative flow plot and quantitative graph. (**B**,**C**) Intratumoral TAM (**B**) and MDSC (**C**) apoptosis representative flow plot indicated with mean ± SD are shown at 4 h post-NPT. (**D**) Changes in CD86 and MHC-II costimulatory marker expression among TAMs were examined on Day 1 or Day 3 post-treatment. A representative flow plot with mean ± SD is shown. *N* = 3–5 per group. Error bars, SD. *: *p* < 0.05 and **: *p* < 0.01 (one-way ANOVA).

**Figure 7 vaccines-12-00633-f007:**
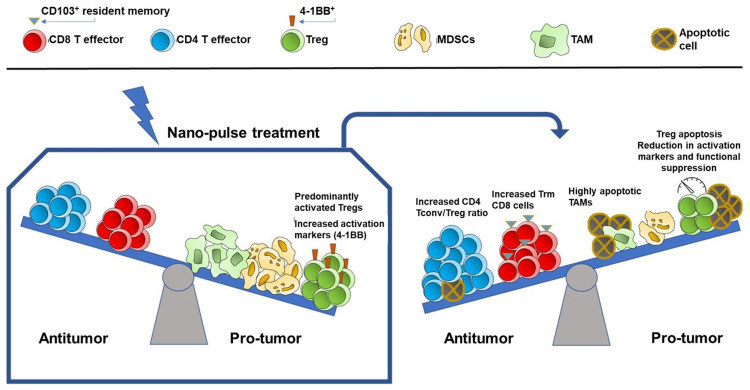
The impact of NPT on the TME in breast cancer. NPT shifts the immune cell dynamics from a pro-tumor to an antitumor environment. Suppressor cells, including Tregs, TAMs, and MDSCs, are diminished. Tregs undergo apoptosis, exhibit a shift from an activated 4-1BB^+^ to a naïve 4-1BB^−^ phenotype, and demonstrate reduced functional suppression capacity post-treatment. Antitumor CD4 Tconv and CD8^+^ CD103^+^ resident memory T cells increase in frequency.

## Data Availability

Data are contained within the article.

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
