# Peer review of "Nano-Pulse Treatment Overcomes the Immunosuppressive Tumor Microenvironment to Elicit In Situ Vaccination Protection against Breast Cancer"

_vaccines, 2024, doi:10.3390/vaccines12060633_

Round 1
Reviewer 1 Report
Comments and Suggestions for Authors
Dear Authors,
Vaccines:
Nano-pulse treatment overcomes immunosuppressive tumor microenvironment to elicit in situ vaccination protection 3 against breast cancer by Nanajian et al demonstrates that nano-pulse treatment (NPT) inhibits 4T1 growth in mouse mammary. This is an interesting article.
Positive comments.
1. This is a novel approach, inhibits tumor growth.
2. Increases immune cells CD43, CD4 and CD8 cells.
3. NPT eradicates activated Tregs and impairs Treg function.
Minor comments:
1. The study used NPT only mouse breast cancer derived 4T1 cells and does it inhibit other mouse cancer derived cancer derived cells.
2. Does NPT work on human cell derived cancer cells.
3. It is ingesting to see broad spectrum of NPT on multiples cancers.
Author Response
Nano-pulse treatment overcomes immunosuppressive tumor microenvironment to elicit in situ vaccination protection 3 against breast cancer by Nanajian et al demonstrates that nano-pulse treatment (NPT) inhibits 4T1 growth in mouse mammary. This is an interesting article.
Positive comments.
- This is a novel approach, inhibits tumor growth.
- Increases immune cells CD43, CD4 and CD8 cells.
- NPT eradicates activated Tregs and impairs Treg function.
Response: We thank the reviewer for these positive comments.
Minor comments:
- The study used NPT only mouse breast cancer derived 4T1 cells and does it inhibit other mouse cancer derived cancer derived cells.
Response: This study was focused on 4T1 mouse breast cancer. However, we previously reported that NPT could treat other cancers and resulted in the changes of immune profiles as well, such as mouse Pan02 pancreatic cancer (Guo, et al. Cancers-2018, Reference # 22) and rat N1S1 liver cancer (Lassiter, et al. Nano-Pulse Stimulation Ablates Orthotopic Rat Hepatocellular Carcinoma and Induces Innate and Adaptive Memory Immune Mechanisms that Prevent Recurrence. Cancers-2018). An early study demonstrated this technology on B16f10 melanoma tumors in mice [Nuccitelli et al., 2006]
- Does NPT work on human cell derived cancer cells.
Response: Since NPT is a form of electrical energy treatment. Like radiation, it should work on many other cancer cells including human cancers as well. So far, studies have demonstrated that it works on mouse 4T1 breast cancer (our group), Pan02 pancreatic cancer (our group), KLN205 lung cancer (our group), B16 melanoma (Nuccitelli and our groups), EL4 lymphoma and CT26 colorectal cancers (Muratori’s group), rat N1S1 liver cancer (our groups), human BxPC-3 pancreatic cancer (Nuccitelli’s group), human basal cell carcinoma in clinical trial (Nuccitelli’s group), etc. In unpublished studies it was effective against MDA-MD 231 human breast cancer.
- It is ingesting to see broad spectrum of NPT on multiples cancers.
Response: Please see the above response. Additionally, we have to emphasize that although NPT can treat many types of cancers including mouse, rat and human, the immune outcomes may differ from various cancer models due to both tumor heterogeneity and their diverse tumor immune microenvironments.
Reviewer 2 Report
Comments and Suggestions for Authors
In their original report, Nanajian and colleagues further investigated their mouse mammary tumor model for in situ vaccination with nano-pulse treatment (NPT) to analyze the underlying immune mechanisms. This model involves the use of established 4T1 breast cancer cells and the orthotopic transplantation of mouse tumor cells into isogenic mice. This process typically results in low immunogenic tumor growth and metastasis. NPT administered on day 11 resulted in tumor-free mice within 7 weeks. Upon re-challenge with tumor cells, most cases (22/27) experienced the rejection of tumor cells, while the remaining cases showed a delay in tumor growth, supporting the idea of a vaccination effect of NPT. Extensive analyses of immune cells support the authors’ conclusions that within their model, NPT can affect immunosuppressive cells in the TME and induce anti-tumor immunogenicity, which they compare with the abscopal effect in tumor radiology. NPT even protected against rechallenge with tumor cells, indicating a vaccine-like effect. This manuscript is informative, well-written, and addresses an important topic relevant to the expected readership of the special issue. Some readers may be interested in understanding the requirements for testing this approach in clinical studies.
Please check Line 259-261 for accuracy/grammar.
Comments on the Quality of English LanguageEnglish is fine.
Author Response
In their original report, Nanajian and colleagues further investigated their mouse mammary tumor model for in situ vaccination with nano-pulse treatment (NPT) to analyze the underlying immune mechanisms. This model involves the use of established 4T1 breast cancer cells and the orthotopic transplantation of mouse tumor cells into isogenic mice. This process typically results in low immunogenic tumor growth and metastasis. NPT administered on day 11 resulted in tumor-free mice within 7 weeks. Upon re-challenge with tumor cells, most cases (22/27) experienced the rejection of tumor cells, while the remaining cases showed a delay in tumor growth, supporting the idea of a vaccination effect of NPT. Extensive analyses of immune cells support the authors’ conclusions that within their model, NPT can affect immunosuppressive cells in the TME and induce anti-tumor immunogenicity, which they compare with the abscopal effect in tumor radiology. NPT even protected against rechallenge with tumor cells, indicating a vaccine-like effect. This manuscript is informative, well-written, and addresses an important topic relevant to the expected readership of the special issue. Some readers may be interested in understanding the requirements for testing this approach in clinical studies.
Please check Line 259-261 for accuracy/grammar:
Response: Thank you for the reviewer’s positive comments. Regarding the readers’ interest in testing this approach in clinical studies, NPT or NPS have been reported to treat human basal cell carcinoma by Nuccitelli’s group, who was motivated to commercialize and translate this approach into clinical therapy for cancer patients. Regarding Line 259-261, “Noticeably, a small percentage (18.5%, 5/27) of animals, which had been treated with NPT but failed to reject the live tumor challenging, gained a prolonged survival with an approximate 10-14 day (Fig. 1B). This is likely because of a slow or delayed tumor growth in these animals at the initial 2-3 weeks following the second tumor challenge.” It was modified to “Noticeably, even though five (18.5%) animals treated with NPT failed to reject the live tumor challenge, these animals still gained an additional 10~14-day survival in comparison to control animals (Fig. 1B). This occurrence is likely due to a slow or delayed tumor growth in these animals during the initial 2-3 weeks following the second tumor challenge.”
Reviewer 3 Report
Comments and Suggestions for Authors
Nano-pulse treatment overcomes immunosuppressive tumor microenvironment to elicit in situ vaccination protection against breast cancer
The present study demonstrates the anti-tumor properties of nano-pulse treatment (NPT) by its immunomodulatory properties on the tumor microenvironment (TME). The approach and the overall design of the study are good. The findings will help potential readers to understand the mechanism of NPT-mediated switch of TME. The first two paragraphs of the discussion should be shortened and included in the introduction section as it is more about the background, and not about the discussion of the present study. It is suggested to include a schematic diagram depicting the overall immunomodulatory action of NPT in TME.
Comments on the Quality of English LanguageMinor editing of the English language is required.
Author Response
The present study demonstrates the anti-tumor properties of nano-pulse treatment (NPT) by its immunomodulatory properties on the tumor microenvironment (TME). The approach and the overall design of the study are good. The findings will help potential readers to understand the mechanism of NPT-mediated switch of TME. The first two paragraphs of the discussion should be shortened and included in the introduction section as it is more about the background, and not about the discussion of the present study. It is suggested to include a schematic diagram depicting the overall immunomodulatory action of NPT in TME.
Response: We agree that the first two paragraphs in the Discussion section should be shortened and included in the introduction. We added a schematic diagram (Fig. 7) in the Discussion section to depict the overall immunomodulation of NPT in the TME. Accordingly, we have modified relevant context as well to suit these changes.